# Evaluations of Deep Learning Approaches for Glaucoma Screening Using Retinal Images from Mobile Device

**DOI:** 10.3390/s22041449

**Published:** 2022-02-14

**Authors:** Alexandre Neto, José Camara, António Cunha

**Affiliations:** 1Escola de Ciências de Tecnologia, University of Trás-os-Montes and Alto Douro, Quinta de Prados, 5001-801 Vila Real, Portugal; alexandre.h.neto@inesctec.pt; 2INESC TEC—Institute for Systems and Computer Engineering, Technology and Science, 4200-465 Porto, Portugal; jcrcamara@hotmail.com; 3Departamento de Ciências e Tecnologia, University Aberta, 1250-100 Lisboa, Portugal

**Keywords:** deep learning, glaucoma screening, retinal images, segmentation, classification

## Abstract

Glaucoma is a silent disease that leads to vision loss or irreversible blindness. Current deep learning methods can help glaucoma screening by extending it to larger populations using retinal images. Low-cost lenses attached to mobile devices can increase the frequency of screening and alert patients earlier for a more thorough evaluation. This work explored and compared the performance of classification and segmentation methods for glaucoma screening with retinal images acquired by both retinography and mobile devices. The goal was to verify the results of these methods and see if similar results could be achieved using images captured by mobile devices. The used classification methods were the Xception, ResNet152 V2 and the Inception ResNet V2 models. The models’ activation maps were produced and analysed to support glaucoma classifier predictions. In clinical practice, glaucoma assessment is commonly based on the cup-to-disc ratio (CDR) criterion, a frequent indicator used by specialists. For this reason, additionally, the U-Net architecture was used with the Inception ResNet V2 and Inception V3 models as the backbone to segment and estimate CDR. For both tasks, the performance of the models reached close to that of state-of-the-art methods, and the classification method applied to a low-quality private dataset illustrates the advantage of using cheaper lenses.

## 1. Introduction

Glaucoma is one of the main causes of vision loss, mainly due to increased fluid pressure and improper drainage of fluid in the eye. In 2013, it was estimated that 64.3 million people aged 40–80 years were diagnosed with glaucoma worldwide. This disease is expected to reach nearly 76 million by 2020 and 111.8 million by 2040. The prevalence of glaucoma is 2.5% for people of all ages and 4.8% for those above 75 years of age [1]. Glaucoma is an asymptomatic condition, and patients do not require medical assistance until a late stage, making the diagnosis frequently too late to prevent blindness. Population-level surveys suggest that only 10–50% of people with glaucoma are aware that they have the disease. As early diagnosis and treatment of the condition can prevent vision loss, glaucoma screening has been tested in numerous studies worldwide [2]. An ophthalmologist can directly examine the eye with an ophthalmoscope or can examine a fundus image capture with a fundus camera, as can be seen in Figure 1. The examination of these fundus images is important because the ophthalmologist can record indicators and parameters related to cupping to detect glaucoma, such as disc diameter, the thickness of the neuroretinal rim (decreasing in the order inferior (I) > superior (S) > nasal (N) > temporal (T) (ISNT rule)), peripapillary atrophy, notching and cup-to-disc ratio (CDR), with this last indicator being the most used measurement by specialists [3,4,5].

Usually, glaucoma is diagnosed on the basis of the patient’s medical history, measures of intraocular pressure (IOP), a visual field loss test and manual evaluation of the optic disc (OD) using ophthalmoscopy to examine the shape and colour of the optic nerve. The examination of the OD is important since glaucoma begins to form a cavity and develops an abnormal depression/excavation at the front of the nerve head, called the optic cup, which, in advanced stages, facilitates the progression of glaucoma, blocking the OD (Figure 2) [6,7,8].

After gathering the retinal images, they must be inspected and analysed to look for indicators of ophthalmologic pathologies. These diagnostic systems offer the potential to be used on a large scale for the early diagnosis and treatment of glaucoma. However, they require subjective evaluation by qualified experts, and it is time-consuming and costly to inspect each retinal image manually. In this regard, deep learning (DL) algorithms help in the automatic screening of glaucoma and assist ophthalmologists in achieving higher accuracy in glaucoma screening, especially in repetitive tasks [3,6,9].

The main objective is to develop a system that allows the screening of low-resolution retinal images captured by a low-cost lens attached to a smartphone. To accomplish this, secondary objectives must be achieved. In this study, state-of-the-art DL methods were explored, tested and applied to high-resolution public databases and then applied to a private database containing low-quality images captured through a low-cost lens attached to a mobile device. These classification methods provide activation maps that allow the model’s decision to be analysed and discussed. Segmentation methods were applied as well, using the CDR to classify images after OD and cup segmentation. These segmentation methods can help an ophthalmologist in a subjective and difficult task, enabling more consistent results that are similar to a clinician’s segmentations. For this purpose, state-of-the-art works on classification and segmentation methods for glaucoma screening were reviewed.

## 2. Literature Review

DL techniques have yielded good results in research on glaucoma screening due to the development of technologies to detect, diagnose and treat glaucoma. The main approach is to conduct screening through computer-aided diagnosis (CAD) systems that use DL to learn and train models through previously labelled available data, identifying patterns and making decisions with minimal human intervention [10]. This section surveys key works with methods using automatic classification models and classification methods using segmentation models.

### 2.1. Classification Methods

The use of classification methods for screening glaucoma lesions in retinal images is another well-established approach. An overview of the best methods recently published is provided in the following.

The study of Gómez-Valverde [2] used the VGG19, GoogLeNet (also known as Inception V1), ResNet50 and DENet models. With these models, Valverde compared the performance between transfer learning and training from scratch. To confirm the performance of VGG19, 10-fold cross-validation (CV) was applied. Valverde used three different databases: RIM-ONE and DRISHTI-GS (public) and Esperanza (private dataset). In the RIM-ONE database, the images classified as suspect were considered to be glaucomatous for the study. The best result was obtained with the VGG19 model using transfer learning.

Diaz-Pinto [11] applied five different ImageNet pre-trained models (VGG16, VGG19, InceptionV3, ResNet50 and Xception) for glaucoma classification and used a 10-fold CV strategy to validate the results. Five databases were used for this work: ACRIMA, HRF, DRISHTI-GS, RIM-ONE and Sjchoi86-HRF. The images were cropped around the OD using a bounding box of 1.5 times the OD radius. All models passed the AUC threshold of 0.96, indicating excellent results.

Serener et al. [12] selected the ResNet50 and GoogLeNet models and trained them with two public databases: a database from Kim’s Eye Hospital (total of 1542 images, including 786 photos from normal patients and 756 from glaucoma patients) and RIM-ONE r3. The database from Kim’s Eye Hospital was used to train the two models, and for the performance evaluation, the models were tested with the RIM-ONE r3 database. With GoogLeNet, Serener obtained better results for early-stage glaucoma than for the advanced glaucoma stage.

The work performed by Norouzifard [13] used two DL models, namely, VGG19 and Inception ResNet V2. These two models were pre-trained and then fine-tuned. For this work, two databases were used: one from the University of California Los Angeles (UCLA) and another publicly available one called high-resolution fundus (HRF). From the UCLA database, they randomly selected 70% of the images for training, 25% for validation and the remaining 5% for testing. To solidify the work, the models were then re-tested with the HRF database. The Inception ResNet V2 model with the UCLA database obtained a specificity and sensitivity above 0.9, even when re-tested with the HRF database.

The study by Sreng [5] was performed in two stages: first, DeepLabv3+ detected and extracted the OD from the entire image, and then three types of convolutional neural networks (CNNs) were used to identify images in the segmented OD region as glaucomatous or normal. After the image was cropped around the OD, 11 ImageNet pre-trained models 23re used: AlexNet, GoogleNet, InceptionV3, Xception, ResNet-50, SqueezeNet, ShuffleNet, MobileNet, DenseNet, InceptionResNet and NasNet-Large. This method was trained with five public databases: REFUGE, ACRIMA, ORIGA, RIM-ONE and DRISHTI-GS. The results showed that DenseNet with the ACRIMA database had the best performance, followed by MoblieNet with the REFUGE database.

### 2.2. Segmentation Methods

Several methods have been published in the literature on segmenting the OD and the cup disc, mostly using adaptations of U-Net. The following presents an overview of the best methods recently published.

Al-Bander [14] proposed a method with a DenseNet incorporated with an FCN with U-shaped architecture. Al-Bander’s approach involved the use of five databases of colour fundus images: ORIGA, DRIONS-DB, DRISHTI-GS, ONHSD and RIM-ONE. For the pre-process, only the green channel of the colour images was considered since the other colour channels contain less useful information. The images were then cropped to isolate the ROI. For OD segmentation, the model achieved better Dice and intersection-over-union (IoU) results with the DRISHTI-GS database compared to RIM-ONE, and the same results were obtained for cup segmentation but with lower values of Dice and IoU compared to OD segmentation.

In the work of Singh [15], a conditional generative adversarial network (cGAN) model was proposed to segment the OD. The cGAN is composed of a generator and a discriminator and can learn statistically invariant features, such as the colour and texture of an input image, and segment the region of interest. For this method, skip connections were used for concatenating the feature maps of a convolutional layer with those resulting from the corresponding deconvolutional layer. To train and evaluate the model, the DRISHTI-GS and RIM-ONE databases were used, with the size of the images reduced to 256 × 256 and the value of each pixel normalised between 0 and 1. For OD segmentation, the model for both databases achieved values above 0.9 for accuracy, Dice and IoU.

Qin [16] proposed neural network constructs utilising the FCN and inception building blocks in GoogleNet. The FCN is the main body of the deep neural network architecture, and to this method, they added several convolution kernels for feature extraction after deconvolution based on the Inception structure in GoogLeNet. Qin’s experiments used two databases: REFUGE and one from the Second Affiliated Hospital of Zhejiang University School of Medicine. For this technique, the authors used a fully automatic method using the Hough circle transform that recognises and cuts the image to obtain an image of the ROI. In the segmentation of the OD and the cup, the model obtained values above 0.9 for Dice and the IoU.

In the work by Yu and others [17], a modified U-Net with a pre-trained ResNet-34 model was developed. This work comprised two steps: first, one single-label modified U-Net model was applied to segment an ROI around the OD, and then after this, the cropped image was used in a multi-label model whose objective was to segment the OD and cup simultaneously. In Yu’s study, the RIGA database was used to train and evaluate the CNN, but then to achieve robust performance, the model trained on RIGA was applied on the DRISHTI-GS and RIM-ONE r3 databases. All of the database images were pre-processed with contrast enhancement, followed by resizing to 512x512 dimensions. In this method, the segmentation of the OD and the cup produced better results with DRISHTI-GS than RIM-ONE r3.

#### Cup-to-Disc Ratio

Glaucoma progression is assessed based on the ratio between OD and cup measurements. The cup-to-disc ratio (CDR) is a clinical method that compares the ratio of the cup to disc, which is currently determined manually, limiting its potential in mass screening. Manual segmentation is dependent on the experience and expertise of the ophthalmologist, so it ends up being subjective and differing between observers [18]. The CDR is commonly used in clinics to classify glaucoma, and specific patterns of change in the region of the OD and cup are used as evidence of glaucoma or glaucoma progression along with other clinical tests, such as intraocular pressure and visual field acuity [18,19].

Accurate segmentation of the OD and cup is essential to a reliable CDR measurement, and reliance on manual effort restricts the deployment of CDR for mass screening, which is fundamental in the detection of early glaucoma for effective medical intervention [18]. Machine learning approaches automatically segment the OD and cup regions and then measure the CDR or extract features that may help to determine whether or not the images contain glaucoma, as can be seen in Figure 3. A higher CDR indicates a higher risk of glaucoma [5,20].

Different parameters can be measured for the CDR to determine the cupping and assess the eye for the presence of glaucoma, such as the horizontal cup diameter to the horizontal OD diameter, the vertical cup disc diameter to the vertical OD diameter and the area of the cup to the area of the OD [19]. If the vertical CDR (VCDR) and horizontal CDR (HCDR) are more than 0.5, the eye is considered to be at risk of abnormality; otherwise, it is considered a normal eye [21]. VCDR and HCDR equations are presented in Equations (1) and (2):(1)$$\mathrm{VCDR}=\frac{V_{\mathrm{cup}}}{V_{\mathrm{disc}}}\;;$$
(2)$$\mathrm{HCDR}=\frac{H_{\mathrm{cup}}}{H_{\mathrm{disc}}}\;.$$

Alternatively, considering the criteria by Diaz [21], the assessment can be performed through the area CDR (ACDR) using a threshold of 0.3, as presented in Equation (3):(3)$$\mathrm{ACDR}=\frac{A_{\mathrm{cup}}}{A_{\mathrm{disc}}}\;.$$

Diaz [21] presented an automatic algorithm that uses several colour spaces and the stochastic watershed transformation to segment the cup and then obtains handcrafted features, such as the VCDR, HCDR and ACDR. Diaz’s method was evaluated on 53 images, obtaining a specificity and sensitivity of 0.81 and 0.87.

After segmentation, Al-Bander [14] calculated the VCDR with varying thresholds and compared the results with an expert’s glaucoma diagnosis, achieving an AUC of 0.74, very close to the 0.79 achieved using ground-truth segmentation. After that, the same approach was used, but with HCDR achieving an AUC of 0.78, close to the 0.77 achieved by the expert’s annotation and higher than the results obtained with the VCDR.

## 3. Materials and Methods

The model pipeline in this work is illustrated in Figure 4. In the first task (Task 1: Data preparation), data pre-processing and organisation processes are described. In the second task (Task 2: Glaucoma screening), different glaucoma classification methods are explained based on classification models alongside the respective activation maps and based on OD and cup segmentation models for CDR calculation. The models and hyper-parameters used for each approach are described in the model setups. In the third and last task (Task 3: Evaluation), the models are evaluated based on each approach’s glaucoma classification.

### 3.1. Data Preparation

Three public databases were used: RIM-ONE r3, DRISHTI-GS and REFUGE. The RIM-ONE r3 database has a balanced proportion between normal and glaucomatous samples, with 85 healthy images and 74 glaucomatous images with a resolution of 2144 × 1424 pixels. The images in this database vary significantly the quality of the illumination and contrast: some are low-light images, making it difficult to identify the OD and cup, and others have good illumination and contrast, helping to identify the retinal components. DRISHTI-GS has a larger representation of glaucoma samples (70 images) than healthy samples (31 images), and the images have a resolution of 2896 × 1944 pixels. Compared to RIM-ONE r3, DRISHTI-GS images have more homogeneous illumination and contrast, which helps to identify and segment the OD and the cup. The REFUGE database is composed of 400 images with a resolution of 2124 × 2056 pixels, but we only had access to the validation set, which has a lower representation of glaucoma samples compared to healthy samples (40 glaucomatous images and 360 normal images).

For each dataset, retina images were divided into a training set (70%), validation set (15%) and test set (15%). The models were trained with each database separately for the segmentation and classification approach. The respective OD and cup masks are available in all of these databases. In the RIM-ONE r3 database, the images classified as suspect were considered glaucomatous, as was also the case in the work by Gómez-Valverde [2]. Since we had little data, the three databases were merged into a larger database (called K-Fold CVDB, standing for K-Fold Cross-Validation DataBase) to perform K-fold cross-validation (CV). The K-Fold CVDB was divided into 5 similar folds for the cross-validation, and one set was left out to test and validate each model and verify the robustness after the training as the final step. The data organisation process is explained in Figure 5.

All images used to train and test the different models were normalised and centralised in the OD and then cropped to focus the CNNs on the ROI. The cropped images have 512 × 512 resolution and did not suffer from changes in illumination or contrast. Augmentation processes were applied to the databases to avoid overfitting the model, such as rotations (range = 0.2), zooms (range = 0.05), shifts (width and heigh shift range = 0.05) and horizontal flips.

### 3.2. Glaucoma Screening

For both approaches, different models were trained with each database separately, and then CV was performed (more precisely, leave-one-out K-fold CV). For this step, the data were partitioned into K equal-sized subsets. K-1 subsets were used to train the CNN, and the remaining set was used for testing. Additionally, the leave-one-out dataset was used for testing the model at the end, giving a more thorough evaluation of each model’s performance since these data were not used to train or test any model. All models were fine-tuned either for image classification or for OD and cup segmentation. Fine-tuning is a procedure based on transfer learning to optimise and minimise the error through the weight initialisation of the convolutional layers using pre-trained CNN weights with the same architecture. The exception is the layer whose number of nodes depends on the number of classes. After the weight initialisation, in the last fully connected layer, the network can be fine-tuned, starting with tuning only the last layer and then tuning the remaining layers, incrementally including more layers in the update process until achieving the desired performance. The early layers learn low-level features, and the late layers learn high-level features specific to the problem in the study [22,23]. For all of the classification and segmentation models used to detect glaucoma, ImageNet pre-trained weights were used. All models selected were based on the best results reported in the reviewed literature.

#### 3.2.1. Classification Methods

Classification used the same principles as segmentation, using pre-trained models with good results inspired by state-of-the-art works. These models were trained with transfer learning using ImageNet weights. First, the four additional layers were pre-trained, freezing the remaining layers before the new ones, and after that, the models were fine-tuned, unfreezing the first layers and training all layers present in the models. We selected the Xcpetion (C1), ResNet 152 V2 (C2) and Inception ResNet V2 (C3) models.

Xception is an extension of the Inception architecture and stands for Extreme Inception. It replaces the standard Inception modules with depthwise separable convolutions called “separable convolution” in frameworks such as TensorFlow and Keras. In the Inception module, filters of different sizes and dimensions are concatenated into a single new filter, acting as a “multi-level feature extractor” by computing 1 × 1, 3 × 3 and 5 × 5 convolutions within the same module of the network. Based on these modules, a more complex and deeper architecture compared to all previous CNN architectures was developed [24]. Depthwise convolution is a spatial convolution performed independently over each channel, followed by a pointwise convolution, i.e., a 1 × 1 convolution. This architecture’s premise is that cross-channel correlations and spatial correlations are sufficiently decoupled to be mapped separately [25].

ResNet is a deep residual network developed with the idea that identifying shortcut connections allows for increasing the depth of convolutional networks while avoiding the gradient degradation problem. These shortcut connections help gradients flow easily in the backpropagation step, which leads to increased accuracy during the training phase. ResNet is composed of 4 blocks with a lot of convolutional blocks inside. Each convolutional operation has the same format in all versions of ResNet (50, 101 and 152), with the only difference being in the number of subsequent convolutional blocks. This deep residual network exploits residual blocks to overcome gradient gradation [23,26].

Inspired by the performance of ResNet, hybrids of Inception and ResNet models were developed. They are two sub-versions of Inception ResNet, i.e., V1 and V2. Inception ResNet V1 has a computational cost similar to that of Inception V3 and Inception ResNet V2 and is similar to Inception v4, with the only difference being in the hyper-parameter settings. They introduce residual connections that use the output of the inception module’s convolution operation as the input of the next module. Therefore, the input and output after convolution must have the same dimensions. To increase depth after convolution, 1x1 convolutions were used after the original convolutions [24].

To train all of these models, images and their respective labels (normal or glaucoma) were used as inputs, and the probability of being one of the classes, normal or glaucoma, was the output.

#### 3.2.2. Segmentation Methods

The availability of a huge dataset such as ImageNet with a high capacity to train the model led to a large variety of pre-trained models for the feature encoder in a CNN. The encoder in a U-Net model is a stack of convolution layers combined with activation functions and pooling layers that can adopt the architecture that is frequently employed for feature extraction with pre-trained models. For the segmentation approach in glaucoma screening, the pre-trained models selected were Inception ResNet V2 and Inception V3 (for simplification, called S1 and S2, respectively). These pre-trained models are used as feature encoders in modified U-Net and use the retina image as input and the respective masks of the OD and cup for training. As output prediction is given, a mask of the OD or cup segmentation is also then used to measure the indicators of glaucoma presence, such as CDR. The predicted mask applies morphological processes to remove holes and anomalies of the prediction if they are present.

#### 3.2.3. Model Setups

**Segmentation models:** The models trained for segmentation were pre-trained for 20 epochs and fine-tuned for 100 epochs with a batch size of 2 for the validation and training sets. The encoder weights were frozen for the pre-training step, and for fine-tuning, the encoder layers were unfrozen; the model was trained again to update all weights. The learning rate started at 10^−4^ with Adam optimiser, and binary cross-entropy was used as the loss function. To prevent the learning rate from stalling on the plateau, the callback reduces the learning rate on the plateau by a factor of 0.90 and only saves the best training weights.

**Classification models:** The classification model was pre-trained for 20 epochs and fine-tuned for 200 epochs with a batch size of 2 for validation and training sets. The learning rate started at 10^−4^ with Adam optimiser, and binary cross-entropy was used as the loss function; to prevent the learning rate from stalling on the plateau, the callback reduces the learning rate on the plateau by a factor of 0.90. All of these models are available in TensorFlow Core and were loaded. The classification layer (last layer/dense layer) was removed, and after that, 4 new layers were added: a global average pooling 2d layer, a dropout layer (dropout = 0.5), a batch normalisation layer and, finally, a dense layer with 2 outputs with SoftMax as the activation function (2 outputs for 2 classes, glaucoma and normal).

### 3.3. Model Evaluation

The metrics for the evaluation of the segmentation model were the intersection over union (IoU) and the Dice coefficient.

The IoU metric measures the accuracy of an object detector applied to a particular database. It measures the common area between the predicted (P) and expected (E) regions, divided by the total area of the two regions, as presented in Equation (4):(4)$$\mathrm{IoU}=\frac{\mathrm{Area}\left(P\cap^{\;}E\right)}{\mathrm{Area}\left(P\cup^{\;}E\right)}\;\;\;\;$$

The Dice coefficient is a statistic used to gauge the similarity between two samples (in this case, between predicted and reference (Ref) segmentation). TP is true positives, FP is false positives and FN is false negatives, as can be seen in Equation (5):(5)$$\mathrm{Dice}=\frac{2\mathrm{TP}}{2\mathrm{TP}+\mathrm{FP}+\mathrm{FN}}\;$$

The CDR equations are described in the previous section in Equations (1)–(3). For the evaluation of the classification models, other metrics were used. The accuracy (Acc) (6) is the fraction of correct predictions by the model.
(6)$$\mathrm{Accuracy}=\frac{\mathrm{TP}+\mathrm{TN}}{\mathrm{TP}+\mathrm{TN}+\mathrm{FP}+\mathrm{FN}}\;$$
where TP is true positives, TN is true negatives, FP is false positives and FN is false negatives. Sensitivity (Sen) (7) measures the proportion of positives that are correctly identified, and specificity (Sep) (8) measures the proportion of negatives that are correctly identified.
(7)$$\mathrm{Sen}=\frac{\mathrm{TP}}{\mathrm{TP}+\mathrm{FN}}\;$$
(8)$$\mathrm{Sep}=\frac{\mathrm{TN}}{\mathrm{TN}+\mathrm{FP}}$$

The F1-score (F1) (9) indicates the balance between precision and recall, where precision is the number of TP divided by the number of all positives, and recall is the number of TP divided by the number of all samples that should have been identified as positive. The F1-score is the harmonic mean of the two.
(9)$$F1\;\mathrm{Score}=\frac{2\mathrm{TP}}{2\mathrm{TP}+\mathrm{FP}+\mathrm{FN}}\;$$

## 4. Results and Discussion

The results are organised in the same way that the methodology is presented in the workflow. For both methods, glaucoma screening was performed by training the models with each database separately and with merged data with the K-Fold CVDB. The results are discussed and compared with the results published by the scientific community. In the end, both methods are compared to assess their capability for glaucoma screening and determine how much they can contribute to supporting this important and challenging task.

### 4.1. Glaucoma Screening Based on Classification Methods

First, for the classification approach, each database was used to train each model separately to determine which model has the best performance and which database has the best quality to produce better model results. The challenge of this methodology is that separately training the models on each database decreases the amount of data that the models learn since there are fewer data to train and validate the training. The results are presented in Table 1.

Overall, the database that showed better results was REFUGE for the C1 and C3 models, with an AUC close to one and with high sensitivity and specificity close to the results presented in Table 2. The C1 and C3 models outperformed those reported by Sreng [5], who also used the REFUGE database for pre-trained networks with a transfer learning model ensemble.

The RIM-ONE and DRISHTI-GS databases showed lower AUC values than REFUGE and the results reported by Sreng [5], who also evaluated each database separately. Better results with a significant difference suggest that the quality of the samples in the REFUGE database is superior to that of the others, with homogeneity in the contrast, illumination and resolution of all samples, in contrast to RIM-ONE and DRISHTI-GS, whose images have heterogeneous quality with a lot of variations in the same factors.

As mentioned in the methodology, the K-fold CV technique was used to train the models with K-1 folds, and then the other test set was used for testing. The K-Fold CVDB was divided into five folds, with an extra set left out to test at the end (leave-one-out method) in all iterations of each model. Of these five folds, four were used to train the model, and the other was used to evaluate it, changing the test fold and training folds in each iteration. The results of the classification in each fold are presented in Table 3.

Compared to the results presented above, when the models were trained with each database separately, the K-fold CV technique showed an immediate enhancement and direct correlation between the amount of data for training and the quality of classification. All of the models showed similar results, with slightly better performance for the C1 model. These models outperformed most of the state-of-the-art works mentioned previously, with some exceptions (Diaz-Pinto [11] and some results of Sreng [5]) owing to fewer data for training. To evaluate the robustness of the models, they were tested with the test set omitted from training, and the results are shown in Table 4.

The results of the classification of the leave-one-out set decreased compared to those discussed above. Nevertheless, most of the models yielded better results than most of the state-of-the-art works, with the same exceptions as those noted previously. The most significant decrease in the results was in the sensitivity, showing a lack of representation and a high rate of false positives for glaucoma samples. The evaluation with the leave-one-out dataset demonstrated that the best model of the three used is C1, as mentioned previously, with the smallest decrease in every metric among all of the models.

The classification models can be a “black-box”: extremely hard to explain and hard for non-experts to understand. Explainable artificial intelligence (AI) approaches are methods and techniques that can explain to humans why the DL models arrived at a specific decision. Explainable AI can create transparency, interpretability and explainability as a foundation for the output of the neural networks. For a visual interpretation of the output to supplement the results of the classification models, activation maps (Figure 6) were created that show the regions of the input images that cause the CNNs to classify the samples as glaucomatous or normal, thus helping clinicians to understand the reason for the output classification.

Gradient-weighted glass activation mapping (Grad-CAM) uses the gradients of any target concept flowing into the final convolutional layer to produce a coarse localization map highlighting the important regions in the image for predicting the concept. These heatmaps can reveal some important indicators or factors for the classification. The used models focussed more on the centre of the OD, where the cupping zone is responsible for and highly correlated with glaucoma cases. The larger the cup area, the more suspicious, and the more probable that the patient has a case of glaucoma. This type of indicator can help ophthalmologists to make a better and more reliable decision, with one of these indicators being the CDR. To calculate this ratio, first, the OD and cup must be segmented, and the trustworthiness of the screening depends on how well they are segmented. The segmentation procedure is time-consuming and inconsistent when performed manually, so to facilitate a more consistent segmentation, we present models for segmentation with a consequent glaucoma classification based on CDR calculation.

### 4.2. Glaucoma Screening Based on Segmentation Methods

The OD and cup were segmented by two different CNNs, and then the different CDRs were calculated; glaucoma was then classified based on the CDR model. This requires the reference (Ref) masks of each database with annotations of the segmentation made by clinicians, and these were available in the databases selected for this work. Finally, the segmentation and glaucoma screening were compared with the reference masks using the same criteria of the glaucoma classification based on CDR values. To perform the segmentation, two different models, S1 and S2, were used. First, the OD segmentation results are presented, followed by the cup segmentation results, and finally, the glaucoma classification based on the CDR calculation with segmentation masks is provided.

#### 4.2.1. OD Segmentation

The procedure in the segmentation methods is the same as the one presented for the classification approach, with the segmentation in each database performed separately and with the K-Fold CVDB. For the K-fold CV, the means of IoU and Dice of the five folds in each model were obtained. The final mask is the intersection/agreement of at least four masks of the five iterations of each model to compute the final CDRs. The results for OD segmentation are presented in Table 5.

At first view, the results in every dataset segmentation are very similar to every compared state-of-the-art method, with a slight but non-significant difference that does not change the outcome of the CDR calculation. This can be explained by the fact that the segmentation of OD is an easy task because of the visible contrast and outline of the OD and the retina, which facilitate identification and segmentation by the neural network. The K-fold CV showed decreases in the IoU and Dice in both models compared to the other results since they represent the mean of five iterations in each model. This can affect the final results, with divergence in the agreement of OD segmentation. However, this difference was not significant enough to jeopardise the CDR calculation, at least in most of the samples. The two models had similar results, with a slightly better performance for S1. After OD segmentation, the procedure was repeated but with different CNN models, this time training the model to segment the cup.

#### 4.2.2. Cup Segmentation

For cup segmentation, the same models were used, but this time, the network was trained to localise and segment the excavation region inside the OD. Contrary to the previous task, cup segmentation is much harder since there is not a high contrast between the exaction zone and the OD (at least not as high as the contrast between the OD and the retina). The results from the two models are presented in Table 6 with the same structure as the one presented for OD segmentation.

Overall, the results achieved the same baseline as the state-of-the-art methods. When directly compared on the RIM-ONE database, the S1 model had better results than S2 and had better Dice than Al-Bander [14], and IoU and Dice were only worse compared to Yu’s [17] work. With DRISHTI-GS, the two models had better IoU and Dice than Al-Bander [14] and a slight difference in IoU and Dice compared to the remaining works, with an overall better performance observed for the S1 model. In the REFUGE database, the results from our models and Qin’s [16] work are very similar, with a minor difference in the Dice, and as observed in the segmentation of the other databases, the S1 model had better results as well.

In the K-fold CV, both models had a major decrease in performance compared to the other works and the performance of the same models using each database separately. As in the previous verification, the S1 model continued to produce better results. Compared to OD segmentation, the IoU and Dice were much lower, which is a consequence of these coefficients being too sensitive to small errors when the segmented object is small and not sensitive enough to large errors when the segmented object is larger.

The results of OD and cup segmentation were used to calculate the CDRs to use as an indicator of glaucoma presence. Reference segmentation by clinicians was used as the ground truth but is not an absolute truth since the segmentation process can be subjective, and the results can differ between clinicians. Thus, the segmentation predicted by the CNNs can sometimes cause the misclassification of the images but can be considered another opinion, especially in cup segmentation since the perimeter of the cup is not as delimitated and visible as the OD. After the segmentation of the OD and cup, the CDRs were calculated to obtain the glaucoma classification.

#### 4.2.3. Glaucoma Screening Based on Estimated CDR

The segmentation masks of the OD and cup of both models were computed and used to calculate the ratio between them. In this work, all CDRs were calculated, including the vertical and horizontal CDRs and the ratio between the areas of the OD and cup. For the VCDR and HCDR, the criteria used were CDR < 0.5 for normal and CDR ≥ 0.5 for glaucomatous, and the ACDR was normal if <0.3 and glaucomatous if ≥0.3, as described in Diaz’s work [21]. The same criteria were used for the Ref masks to allow a direct comparison between the results of our models and the segmentation performed by ophthalmologists to gauge the reliability of segmentation by the S1 and S2 models. The results are expressed in Table 7.

The results for both models were similar to the results using the Ref masks, which indicates that they produced similar segmentation results or at least provided similar CDRs. Overall, the results from CDRs based on the Ref masks were better than the results from the two models, but the difference between the models’ classification and the classification in the Ref masks, in a lot of cases, was not significant.

With RIM-ONE, the Ref had a better F1-score for the VCDR and HCDR, but the difference in the F1-scores between the two models was very small. S2 achieved better sensitivity and specificity, but this difference was also small, which may indicate that the masks were very close to each other or had similar forms that led to the computation of similar CDR values. DRISHTI-GS and the K-Fold CVDB were the two datasets with the worst results for both models in comparison with the Ref results, showing a greater difference, but the AUC indicated that the difference was not that large. The results from REFUGE were better for the S2 model compared to S1 and Ref for sensitivity, specificity and F1-score, but in all models and the Ref, the values were very low, which may suggest that, in this case, the CDRs are not a sufficient indicator to produce a classification of glaucoma or normal; thus, for a better decision, complementary information is needed to support the final call. All of the ROC curves from the different databases for all CDRs of the models and Ref masks are presented in Figure 7.

Of all CDRs, the VCDR and ACDR had the best results. The HCDR was the worst result in the two models and the Ref, and the model with the overall best results was S2. This is also shown in the ROC curves, with the models and the Ref having very similar results in all AUCs for the glaucoma classification based on the different CDRs. The difference between the AUCs of the models and the Ref was not significant and was generally very small, with the Ref showing slightly better performance than the S2 model. This can reinforce the notion that the masks originating from the S1 and S2 models are very close to the Ref masks or compute similar CDRs that lead to a similar glaucoma classification based on CDRs. In the work by Diaz [21], the model obtained specificity of 0.81 and sensitivity of 0.87, and Al-Bander [14] achieved an AUC of 0.74 using the VCDR and 0.78 using the HCDR. The majority of our results surpass the results of the state-of-the-art glaucoma classification methods based on CDRs.

For the K-fold CV, the results of the ROC curves for both models were very similar to those of the Ref and had close AUC values, except for the HCDR. As mentioned previously, the HCDR was the CDR that differed the most, as can be seen in Figure 8.

For visual comparison, in the following images, the masks and outlines of both models are drawn in red, and those of the Ref masks are drawn in green. The intersection between the masks predicted by the models and the Ref masks is indicated by the combination of green and red (true positive). The green area represents a false negative since there is no intersection between the masks, and the red area represents a false positive since the model’s prediction does not correspond to the same result as the Ref mask.

In Figure 9, the CDRs values for higher Dice cases are extremely close to the Ref CDR values. In the K-fold, the resulting masks are the intersection of at least four agreements in the model of the different folds. The predicted masks of the OD and cup are very close to the Ref masks, reflecting high IoU and Dice values. In the lower Dice cases, the CDRs significantly differ compared to Ref CDRs, but despite this, there is complete agreement in the final decision for the classification based on CDRs since all apply the same threshold values, although they differ more than the higher Dice cases.

The two models used achieved state-of-the-art results for the segmentation, and the outcome was similar to the glaucoma classification based on the CDR with the Ref masks, indicating that these types of models can mitigate these labour-intensive and subjective tasks, that is, the segmentation of the OD and cup, providing a more consistent final result. To complement the CDR indicator, additional examination must be performed to make the final diagnosis of the patient using, for example, IOP values, anamnesis data and medical records. Another problem is the thin margin in the threshold CDRs, potentially resulting in an arbitrary classification; to resolve this obstacle, more diagnosis classes can be added based on CDRs, such as a suspicious case of glaucoma in the samples for which the CDR value barely passes or reaches the threshold.

### 4.3. Classification Methods on a Private Dataset

For the glaucoma screening based on DL methods, only classification models were applied on the private dataset, since this did not have ground-truth masks for the application of segmentation techniques. For the classification, the same K-fold CV approach was applied to a private dataset of D-EYE (Portable Retinal Imaging System) images with lower resolution. The goal was to see if applying the classification methods to images acquired by mobile devices could achieve similar results to those obtained using high-resolution images captured by clinical equipment. This would impart some portability to the glaucoma screening process, expanding it to more people and preventing glaucoma cases. This dataset was approved by the Ethical Committee of the Universidade Aberta of Lisbon and by the Health Ministerium of Brazil following the dispatch of the information DW/2018 of 02-21-2019 provided by the Brazilian Research Ethics Committee. This dataset consists of D-EYE images collected between October 2018 and March 2020 from patients aged above 40, either treated or untreated for glaucoma; subjects accepted the research protocols and allowed the use of data for studies on applications of automatised methods of glaucoma screening.

The images were obtained using a lens of D-EYE coupled to the camera of an iPhone 6S, which allows photographing the patient’s optical papilla through 75 and 90 diopters and recording a short video that is stored in .mp4 format and collected in an environment with low light. From the videos, images that had 1080 × 1920 pixels of resolution and underwent the same pre-processing treatment as described for the glaucoma screening with the public databases were selected, and the OD was cropped and centred, obtaining dimensions of 512 × 512. From the database, a total of 347 images were selected, of which 293 were classified as normal, and 54 were classified as glaucomatous.

For the classification, since it is a small database, K-fold CV was applied with a leave-one-out set to classification neural networks. Each database was divided into five folds, in which each fold had 49 normal samples and 9 glaucomatous samples. The leave-one-out set had 48 normal samples and 9 glaucomatous ones and was used to validate the models after training. To train the CNNs, the same pre-trained classification models were used, namely, C1, C2 and C3. The results are presented in Table 8.

The models obtained high AUC values and specificity but low sensitivity and F1-score results, showing that they had difficulty classifying the glaucomatous samples since the database lacks sufficient representation of glaucoma samples, and most of these images were classified based on the clinical record, family history and IOP values. The glaucoma images do not show consistent patterns that indicate glaucoma incidence directly in the image. To validate the model’s performance, each one was tested with the leave-one-out dataset, and the results are presented in Table 9.

In Figure 10, image Figure 10a illustrates a case of glaucoma with cataract opacity that worsens the overall quality of the image. Nevertheless, the C3 model predicted the sample correctly. The activation map points to a peripheric region with the presence of vases, but the spot that indicates the incidence of glaucoma is located in the centre of the excavation zone. Other good examples of a poor focus on the region of interest are images Figure 10b,c, where the output prediction was correctly classified, but the activation map points to an excentric zone instead of focussing on the cup area.

Figure 10c shows slightly better recognition of the centre zone, but the models focus on the veins and not on the optic cup. The CNNs provide additional information for diagnosis by an ophthalmologist but, in most cases, have an excentric focus (on veins) instead of focussing on the centre area (cupping of the OD).

The CNNs can help clinicians to expand glaucoma screening and accelerate the early screening of glaucoma. This database lacks visual representation of glaucoma samples since most of the glaucoma images collected do not have visual signs of glaucoma, and the diagnoses were made based on other indications, such as IOP, clinical records and family history, as mentioned previously. To improve the results, a more balanced database is needed with more glaucoma samples with visual patterns and indicators that evidence the presence of glaucoma. Another way to obtain better results would be to use other types of clinical data in neural networks to complement the image data.

## 5. Conclusions

Glaucoma disease has a high incidence around the world, the main cause of which is the lack of tools for and accessibility to early screening to prevent the evolution of the disease. Since it is a disorder that is usually asymptomatic, it is frequently detected in the late stage; by this time, medical treatment cannot reverse the injuries and vision loss but can only prevent the spread of glaucoma. Glaucoma screening is carried out in clinical centres by specialised clinicians with expensive tools. Mass screening is time-consuming and, most of the time, subjective, especially in the early stage, depending on the expertise of the ophthalmologist. For this reason, different approaches using CNNs can help to expand mass glaucoma screening, save time and money and help medical staff to perform more reliable screening with more consistent decisions, speeding up the process and relieving hard and repetitive work.

All classification models achieved results similar to those of state-of-the-art methods, with the Xception model showing an overall better performance. The CNN models for classification, unlike the CDR and segmentation method, are “black-boxes”; they do not provide a visual representation of their decisions. Thus, in this work, the model’s activation maps are presented to provide visual interpretations and analyse the model’s classification, thus helping medical experts to understand the CNN’s decision. A careful analysis reveals that, in this case, the CNNs focus on the centre of the OD in the cup, reinforcing the significance of the cupping area as an indicator of glaucoma presence. The OD and cup can provide the CDR, which is usually used as an indicator for glaucoma screening. In other cases, the activation maps focus on peripheral veins that, in most cases, do not correlate with the incidence of glaucoma.

Since the ratio between the OD and cup is the most used indicator in the ophthalmology field, segmentation methods were applied to classify the samples after classification based on CDRs. The segmentation of the cup is more difficult than segmenting the OD, which does not usually have a well-defined boundary to help in the segmentation, making it difficult for clinicians to perform this task. For this reason, the CNNs have proved to be helpful in facilitating a subjective and hard task that highly depends on the experience of the ophthalmologist. The CDRs computed through the segmented masks were very close to the Ref CDRs, reinforcing that the CNNs can conduct an evaluation similar to that performed by a clinician. The model that produced the best results overall for these tasks was Inception V3 as the backbone of U-Net, with slightly better performance for the different CDRs. A way to improve classification based on CDR calculations is to use an additional class instead of binary classification, providing an extra margin for the threshold.

The classification methods were applied to a private database with images collected through a lens attached to a mobile device, and the results are promising since this lens is cheaper and can expand the accessibility and accelerate mass glaucoma screening. The model with the best results in the private database was Inception ResNet V2, which had higher sensitivity compared to the remaining models. The Xception model achieved similar AUC results but had a lower sensitivity compared to Inception ResNet V2. The classification results of images in this private database are promising but did not achieve the sensitivity of the models trained with public databases. The model’s classification can facilitate mass screening with images collected by lenses attached to mobile devices, serving as an extra opinion and providing activation maps to explain the model’s decision. These new approaches of collecting retinal images with posterior CNN classification models can accelerate and contribute to mass screening, mostly in remote areas, helping to redirect people to medical centres to prevent glaucoma as early as possible.

## Figures and Tables

**Figure 1 sensors-22-01449-f001:**
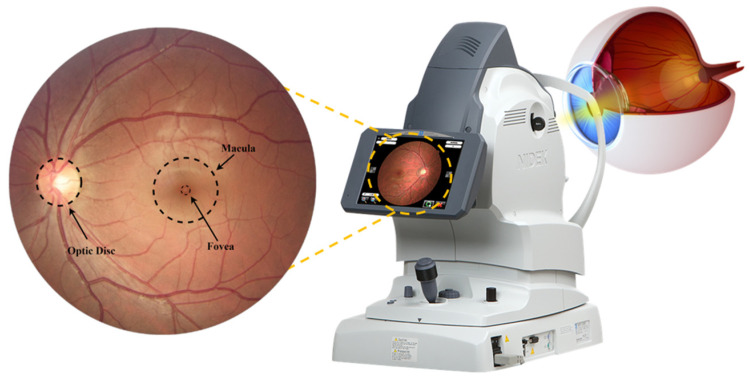
Representation of capturing an image of the interior surface of the eye (retina).

**Figure 2 sensors-22-01449-f002:**
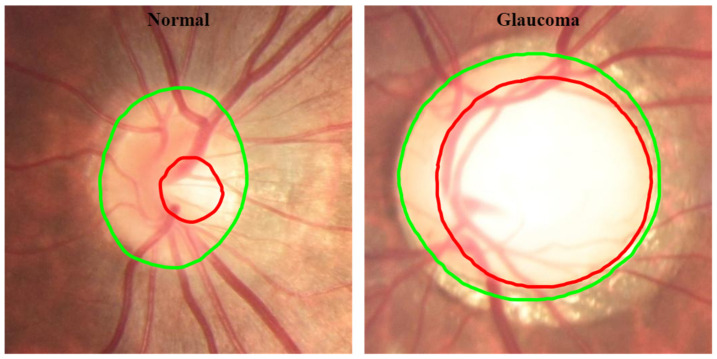
Retinal image from the normal and glaucomatous eye. **Green line:** OD boundary; **red line:** cup boundary.

**Figure 3 sensors-22-01449-f003:**
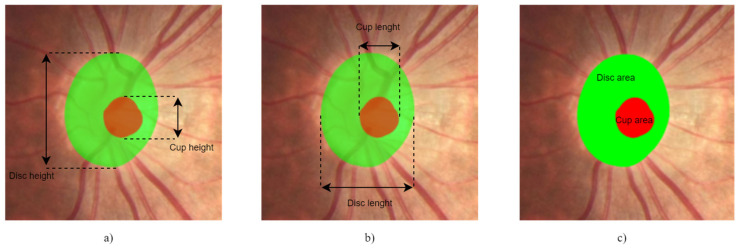
CDR: (**a**) VCDR; (**b**) HCDR; (**c**) ACDR.

**Figure 4 sensors-22-01449-f004:**
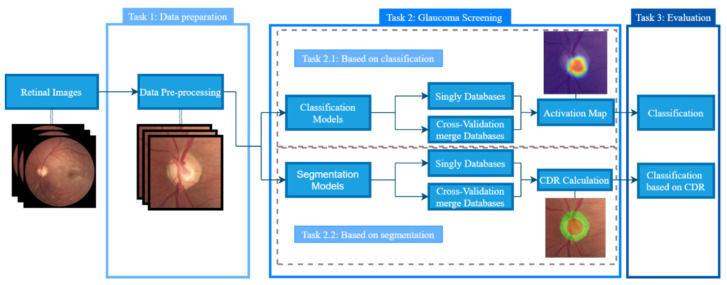
The model pipeline for glaucoma screening.

**Figure 5 sensors-22-01449-f005:**
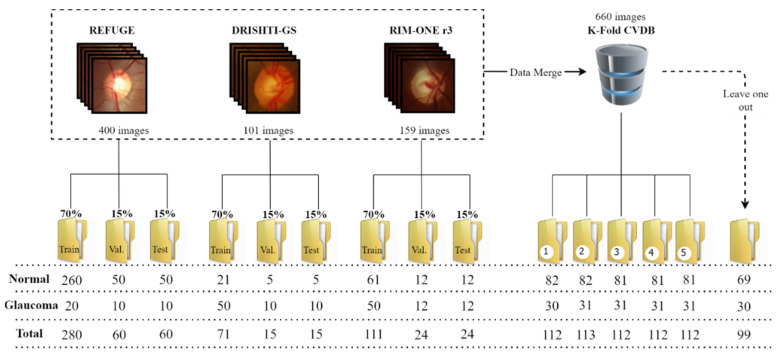
Data organisation for training the models.

**Figure 6 sensors-22-01449-f006:**
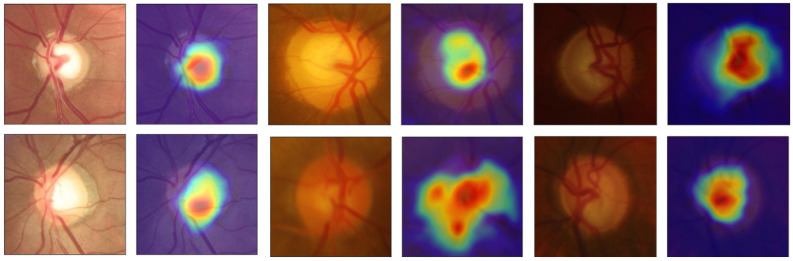
Activation maps of the classification models of the K-fold CV (left, original; right, heatmap). The importance is indicated by the emphasized shades in the following order, from the most important to the least important: red, orange, yellow, green and blue.

**Figure 7 sensors-22-01449-f007:**
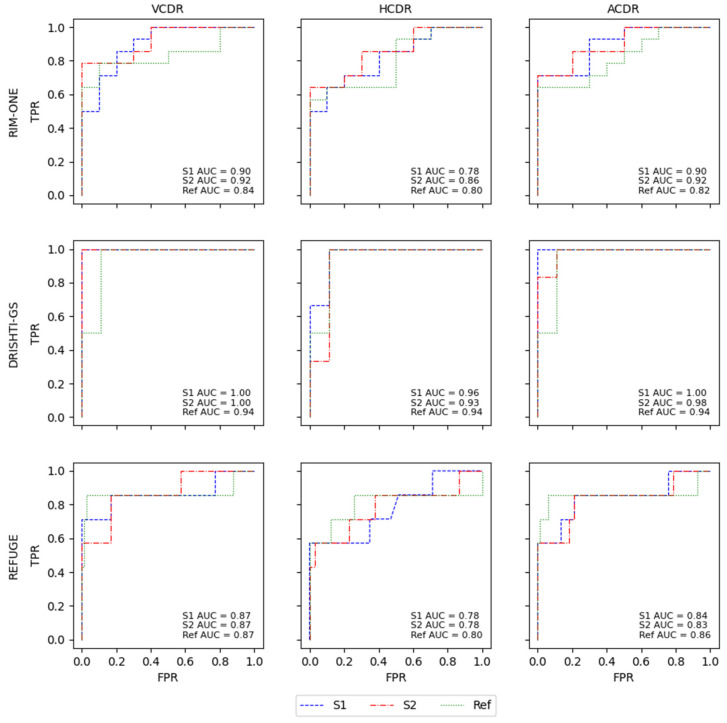
ROC curves for the glaucoma classification through CDR calculations for each database separately with the S1 and S2 models and the respective CDR calculations with the Ref masks.

**Figure 8 sensors-22-01449-f008:**
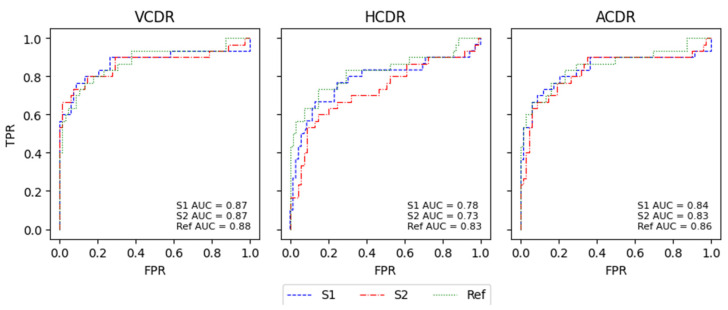
ROC curves for the glaucoma classification through CDR calculation for K-Fold CVDB with the S1 and S2 models and with the respective CDR calculations with the Ref masks.

**Figure 9 sensors-22-01449-f009:**
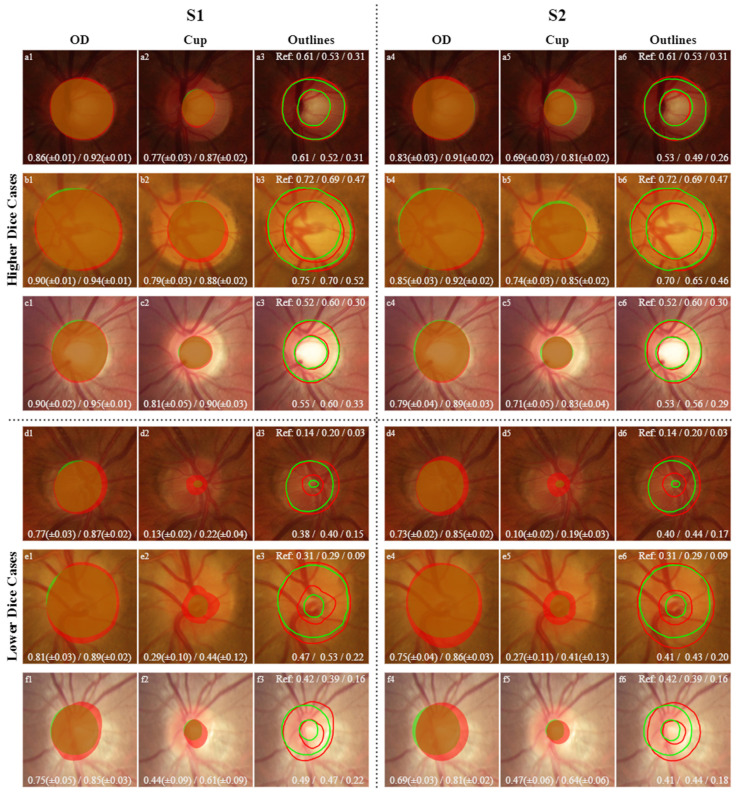
Higher Dice cases and lower Dice cases for each model for OD and cup segmentation with the final outlines of the OD and cup for the K-Fold CVDB. Green represents the masks and outlines of the Ref, and red represents the masks and outlines of the predictions from the models. The lower right corner shows the mean IoU/Dice for OD and cup segmentation for every fold in each model with the respective standard deviation; the “Outlines” columns at the top are the Ref values of the CDRs (VCDR/HCDR/ACDR), and the bottom of the images show the results from CDRs of the models in the same order as that described for the Ref.

**Figure 10 sensors-22-01449-f010:**
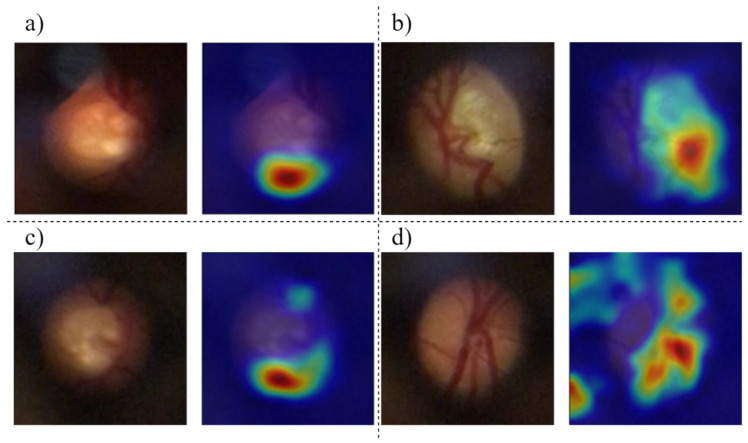
Activation maps for D-EYE images of the CV models. The importance is indicated by the emphasized shades in the following order, from the most important to the least important: red, orange, yellow, green, blue. All images report glaucoma cases (**a**–**d**).

**Table 1 sensors-22-01449-t001:** Results for the models trained separately on each database.

Database	Model	Acc	Sen	Spe	AUC	F1
RIM-ONE	C1	0.83	0.79	0.90	0.84	0.85
	C2	0.63	0.64	0.60	0.75	0.67
	C3	0.67	0.79	0.50	0.70	0.73
DRISHTI-GS	C1	0.67	0.67	0.67	0.80	0.71
	C2	0.60	0.89	0.17	0.72	0.73
	C3	0.73	0.78	0.67	0.79	0.79
REFUGE	C1	0.97	0.83	0.98	0.98	0.83
	C2	0.85	0.67	0.87	0.74	0.47
	C3	0.95	0.83	0.96	0.99	0.77

**Table 2 sensors-22-01449-t002:** Results of state-of-the-art glaucoma classification methods.

Author	Model	Database	Sen	Spe	AUC
Gómez-Valverde [2]	VGG19	RIM-ONE, DRISHTI-GS and Esperanza	0.84	0.89	0.92
	GoogLeNet		0.89	0.83	0.93
	ResNet50		0.84	0.89	0.92
	DENet		0.81	0.88	0.91
Diaz-Pinto [11]	VGG16	ACRIMA, HRF, DRISHTI-GS, RIM-ONE and Sjchoi86-HRF	0.91	0.88	0.96
	VGG19		0.92	0.88	0.96
	Inception V3		0.92	0.88	0.97
	ResNet50		0.90	0.89	0.96
	Xception		0.93	0.86	0.96
Norouzifard [13]	Inception ResNet V2	UCLA and HRF	0.91	0.93	-
Sreng [5]	Pre-train CNNs as feature extractors models ensemble	RIM-ONE	-	-	0.99
		DRISTHI-GS	-	-	0.92
		REFUGE	-	-	0.94
		ORIGA	-	-	0.85
		ACRIMA	-	-	0.99

**Table 3 sensors-22-01449-t003:** Results for the models with K-fold CV for each test set of each fold with the mean results of the models for the 5 folds and the standard deviation.

Model	Acc	Sen	Spe	AUC	F1-Score
C1	0.91 (±0.04)	0.86 (±0.10)	0.93 (±0.04)	0.96 (±0.02)	0.84 (±0.07)
C2	0.90 (±0.02)	0.81 (±0.08)	0.93 (±0.02)	0.94 (±0.02)	0.81 (±0.05)
C3	0.88 (±0.03)	0.77 (±0.10)	0.92 (±0.04)	0.94 (±0.02)	0.78 (±0.05)

**Table 4 sensors-22-01449-t004:** Results for the models in K-fold CV for the leave-one-out test set with the respective mean and standard deviation for the 5 folds of each model.

Model	Acc	Sen	Spe	AUC	F1-Score
C1	0.84 (±0.01)	0.78 (±0.03)	0.87 (±0.01)	0.91 (±0.01)	0.75 (±0.02)
C2	0.82 (±0.03)	0.69 (±0.10)	0.87 (±0.02)	0.87 (±0.03)	0.69 (±0.07)
C3	0.80 (±0.02)	0.65 (±0.10)	0.86 (±0.04)	0.86 (±0.04)	0.66 (±0.05)

**Table 5 sensors-22-01449-t005:** Results for OD segmentation for each model and the K-Fold CVDB. For comparison, the results from the literature review are presented as well. S1 is Inception ResNet V2, and S2 is Inception V3.

Database	Model	IoU	Dice
RIM-ONE	S1	0.91 (±0.08)	0.95 (±0.05)
	S2	0.89 (±0.05)	0.94 (±0.03)
	Al-Bander [14]	0.83	0.90
	Singh [15]	0.93	0.98
	Yu [17]	0.93	0.96
DRISHTI-GS	S1	0.94 (±0.02)	0.97 (±0.01)
	S2	0.93 (±0.02)	0.96 (±0.01)
	Al-Bander [14]	0.90	0.95
	Singh [15]	0.96	0.97
	Yu [17]	0.95	0.97
REFUGE	S1	0.93 (±0.03)	0.96 (±0.02)
	S2	0.92 (±0.03)	0.96 (±0.02)
	Qin [16]	0.92	0.97
K-Fold CVDB	S1	0.83 (±0.01)	0.91 (±0.01)
	S2	0.81 (±0.01)	0.89 (±0.004)

**Table 6 sensors-22-01449-t006:** Results for cup segmentation for each model and the K-Fold CVDB. For comparison, the results from the literature review are represented as well.

Database	Model	IoU	Dice
RIM-ONE	S1	0.70 (±0.13)	0.82 (±0.09)
	S2	0.68 (±0.15)	0.80 (±0.12)
	Al-Bander [14]	0.56	0.69
	Yu [17]	0.74	0.84
DRISHTI-GS	S1	0.76 (±0.22)	0.84 (±0.19)
	S2	0.74 (±0.20)	0.83 (±0.19)
	Al-Bander [14]	0.70	0.82
	Yu [17]	0.80	0.89
REFUGE	S1	0.81 (±0.07)	0.90 (±0.04)
	S2	0.80 (±0.06)	0.89 (±0.04)
	Qin [16]	0.90	0.92
K-Fold CVDB	S1	0.64 (±0.03)	0.77 (±0.02)
	S2	0.60 (±0.02)	0.74 (±0.02)

**Table 7 sensors-22-01449-t007:** Results of glaucoma classification with CDR calculations for S1, S2 and Ref masks.

		S1				S2				Ref			
		Sen	Spe	F1	AUC	Sen	Spe	F1	AUC	Sen	Spe	F1	AUC
**RIM-ONE**	**VCDR**	0.71	0.80	0.77	0.90	0.79	0.90	0.85	0.92	0.75	1.00	0.86	0.84
	**HCDR**	0.71	0.70	0.74	0.78	0.64	0.90	0.75	0.86	0.75	0.86	0.80	0.80
	**ACDR**	0.71	0.90	0.80	0.90	0.64	1.00	0.78	0.92	0.50	1.00	0.67	0.82
**DRISHTI-GS**	**VCDR**	1.00	0.17	0.78	1.00	1.00	0.50	0.86	1.00	1.00	0.67	0.90	0.94
	**HCDR**	1.00	0.20	0.82	0.96	1.00	0.33	0.82	0.93	1.00	0.67	0.90	0.94
	**ACDR**	1.00	0.50	0.86	1.00	1.00	0.50	0.86	0.98	1.00	0.67	0.90	0.94
**REFUGE**	**VCDR**	0.80	0.67	0.30	0.87	0.80	0.73	0.33	0.87	1.00	0.72	0.33	0.87
	**HCDR**	0.80	0.36	0.18	0.78	0.80	0.44	0.20	0.78	1.00	0.43	0.20	0.80
	**ACDR**	0.60	0.82	0.33	0.84	0.80	0.85	0.47	0.84	1.00	0.79	0.40	0.86
**K-Fold CVDB**	**VCDR**	0.90	0.68	0.68	0.87	0.90	0.70	0.69	0.87	1.00	0.60	0.91	0.88
	**HCDR**	0.83	0.30	0.49	0.78	0.80	0.46	0.53	0.73	1.00	0.40	0.87	0.83
	**ACDR**	0.80	0.78	0.70	0.84	0.70	0.84	0.68	0.83	0.90	0.60	0.86	0.86

**Table 8 sensors-22-01449-t008:** Results for the models with K-fold CV for each test set of each fold with the mean of results of the models for the 5 folds and the standard deviation.

Model	Acc	Sen	Spe	AUC	F1-Score
C1	0.87 (±0.03)	0.36 (±0.11)	0.96 (±0.03)	0.82 (±0.05)	0.45 (±0.10)
C2	0.86 (±0.02)	0.16 (±0.11)	0.99 (±0.02)	0.84 (±0.06)	0.24 (±0.16)
C3	0.87 (±0.05)	0.47 (±0.15)	0.94 (±0.07)	0.87 (±0.05)	0.52 (±0.14)

**Table 9 sensors-22-01449-t009:** Results for the models in K-fold CV for the leave-one-out test set with the respective mean and standard deviation for the 5 folds of each model for the private dataset.

Model	Acc	Sen	Spe	AUC	F1-Score
C1	0.85 (±0.02)	0.36 (±0.16)	0.95 (±0.01)	0.79 (±0.04)	0.42 (±0.14)
C2	0.85 (±0.04)	0.29 (±0.05)	0.95 (±0.05)	0.77 (±0.22)	0.38 (±0.07)
C3	0.85 (±0.04)	0.49 (±0.22)	0.92 (±0.08)	0.80 (±0.07)	0.49 (±0.10)

## Data Availability

Not applicable.

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
