# Peer review of "Evaluations of Deep Learning Approaches for Glaucoma Screening Using Retinal Images from Mobile Device"

_sensors, 2022, doi:10.3390/s22041449_

Round 1

Reviewer 1 Report

The quality of reserch is high. Here is clearly presented all research results. 

Author Response

Thank you for your valuable comments.

Reviewer 2 Report

This paper explores and compares the contributions of classification and segmentation CNNs methods for automatic glaucoma screening with retinal images acquired both by retinographers and a mobile device. The methods and data analyses are well described. The results are solid. Overall, I suggest minor revisions from authors. Here are my comments.

  1. As mentioned in the introduction, the examination of the OD is important since glaucoma begins to form a cavity and develops an abnormal depression/excavation. What’s are the key points for diagnosis the disease based on such image data? And why the deep learning glaucoma screening can enhance the conventional methods?

Reviewer 3 Report

*The title of the manuscript is not accurate.  As the title has the term “segmentation vs classification”, at first, I was thinking that this paper will compare segmentation approaches with classification approaches, and evaluate which approaches are better in glaucoma screening.  However, this paper does not have such comparisons.  Therefore, in my opinion, it would be better if the authors could modify the title of this paper.

*All abbreviations should be defined.  For example, what is ISNT in the abstract?

*There are grammatical errors and typos in this paper.  Please proofread before re-submission.

*In line 7, it is stated that the method is “Xcpetion”, but in line 270, it is “Exception”.  However, in my opinion, they should be “Xception”.

*There is no novelty in terms of the design of deep learning network, as all of the networks are proposed by other researchers.  However, the novelty of the work may be in terms of the development of a diagnosis device using mobile device.  Thus, in my opinion, the author should emphasize more on this part in this paper.  Currently, it is only covered in Section 4.3.  For example, the author could also show a figure of the device they developed. 

*In line 224.  It is “RIM-ONE” or “RIM-ONE r3”?

*For Section 3.2.1 and 3.2.2., it is not clear why the authors select Exception (C1), ResNet 152 V2 (C2), Inception ResNet V2 (C3) , Inception ResNet V2 (S1) and Inception V3 (S2)?  Why not other methods?  Besides, some of the methods are already been used by other researchers for glaucoma classification.  For example, Xception has been used by Diaz-Pinto [11].  Thus, it is not clear on the actual aim of this research.  Thus, the authors should emphasize more on their research aims. 

*Besides, Table 2 already listed some of the results to compare the performance of the classification methods.  Thus, why we need to do other evaluations, as given in Table 1?

*It would be better if the authors could provide some more descriptions on how the activation maps, as shown in Figure 6 and Figure 10, are generated.  Besides, it would be better if a legend is given to show the color scale.

*In line 415, it is stated that the reference masks are made by the clinicians.  Is this means that the datasets come with the reference masks, or the authors find the clinicians to create these masks?  Please clarify this in text.

*In line 558, “thesis” should be “paper”.

*In the first paragraph of Section 4.3, please provide more details about the ethical approval, such as the approval number and the body that gave the approval.

Reviewer 4 Report

General Comments:

The paper presents an approach for glaucoma diagnosis via deep learning. The topic is important for retinal image research, and the proposed approach is interesting. However, the contribution versus literature is not clear.

Specific Comments:

  1. The Abstract should explain the main contribution of this work. Specifically, to what extent a low-resolution retinal image can give accurate results.
  2. Section 3.1: The augmentation process is unclear.
  3. Section 3.2: The structure of the proposed CNN should be analyzed and justified. Segmentation and classification processes should be explained in detail, showing the contribution of this work versus existing works.

Language usage:

The paper should undergo moderate language revision. Just as examples:

  1. Line 19, “In the clinical” should be “In the clinical practice”.
  2. Line 19, “criterium” should be “criterion”.
  3. Line 22, “reached performances near the state-of-the-art” should be “achieved performance that is near the state of the art” or “reached state-of-the-art performance”. Note that state-of-the-art (with dashes) is an adjective.
  4. Line 23, “show contribution for the use of cheaper lens” should be “provides the advantage of using cheaper lens”.
  5. Line 28, “In 2013, was estimated that” should be “In 2013, it was estimated that”.
  6. Lines 100-101, The structures including “was selected” and “training them” should be amended.
  7. Line 217, a sentence cannot start with “Was”.

… etc

Round 2

Reviewer 3 Report

Dear authors,

Thanks for addressing most of my previous concerns.  Yet, unfortunately, I found that some parts of the paper still need further clarification/improvement:

1) There are still writing mistakes in this paper.  For example, "Xcpetion" is still not been corrected in the abstract (line 17).  Some sentences have grammatical errors, such as in line 101, "In Serener and Serte study [12] were selected ...". 

2) In line 597, I think "The goal is to see if the classification methods used in high-quality images captured by clinical equipment can achieve similar results using images acquired by mobile devices." should be "The goal is to see if the classification methods used in images acquired by mobile devices  can achieve similar results as using high-quality images captured by clinical equipment."

3) Section 4.1 gives the results from classification methods, whereas Section 4.2 gives the results from segmentation approaches.  However, for Section 4.3, only classification approaches are investigated.  Why not also includes the segmentation approaches?  Or, alternatively, at the beginning of Section 4.3, add one paragraph to justify why this section only use classification approaches (based on the findings from Section 4.1 and Section 4.2).

4) What is "D-EYE" stands for?

5) On line 596, the authors state about "lower resolution".  The resolution here refers to the spatial resolution, or bits-per-pixel?  If it is referring to the spatial resolution, it would be better if in Section 3.1, the spatial resolutions of the images in RIM-ONE r3, DRISHTI-GS and REFUGE are provided.  And in Section 4.3, the spatial resolution of D-EYE images are provided.

6) In line 597, the paper states about "high-quality images".  What does this quality refers to?  Is it referring to the contrast, noise, or other attribute?  In my opinion, better if some measure be provided, to show the difference of quality between those three databases, with D-EYE images.  For example, the authors could provide the average contrast value for the database, and the average contrast value for the D-EYE images.

7) The quality of the images are also depending on the lens and the image sensors (the mobile phone).  Thus, better to provide specific model information in Section 4.3.  Are these images captured by one mobile phone, or by many phones?

8) The current title indicates that the authors are suggesting a new deep learning method. The term "retinography" is not been used in the text. As the authors comparing the available methods, I suggesting the following title for consideration:

"Evaluations of deep learning approaches for glaucoma screening using retinal images from mobile device".
